# Improving the Dispersion Behavior of Organic Components in Water-Based Electrode Dispersions for Inkjet Printing Processes

**Cara G. Kolb** \*, **Maja Lehmann**, **Jana-Lorena Lindemann, Andreas Bachmann** and **Michael F. Zaeh**

Institute for Machine Tools and Industrial Management, Department of Mechanical Engineering, Technical University of Munich, Boltzmannstr. 15, 85748 Garching, Germany; maja.lehmann@iwb.tum.de (M.L.); jana-lorena.lindemann@tum.de (J.-L.L.); andreas.bachmann@iwb.tum.de (A.B.); michael.zaeh@iwb.tum.de (M.F.Z.)
\* Correspondence: cara.kolb@iwb.tum.de; Tel.: +49-89-289-15589

**Abstract:** Water-based processing of electrodes is associated with an enhanced environmental footprint for lithium-ion battery (LIB) production in conjunction with reduced costs. This trend is accompanied by an increasing demand for electrode dispersion processing in inkjet printing. However, most of the dispersion components show a low inherent dispersibility with poor stability in aqueous formulations. This is particularly important when it comes to qualifying electrode dispersions for use in inkjet printing, since the effect of agglomeration and sedimentation effects must be effectively prevented. Therefore, additives are needed to improve the dispersive behavior. This paper analyzes the suitability of dispersants for organic electrode components, in particular graphite and carbon black. An empirical approach was devised on the basis of comprehensive theoretical considerations. Empirical investigations revealed that the utilization of polyvinylpyrrolidone (PVP) favored the enhanced stabilization of graphite nanoparticles. The addition of Triton X-100 (TX-100) resulted in an improved stabilization of carbon black. Based on these empirical studies, a methodology was derived, which supports the application of suitable dispersants in printable dispersions.

**Keywords:** lithium-ion batteries; Inkjet printing; electrode processing; water-based slurries; stability behavior; additives; dispersants

## 1. Introduction

Lithium-ion batteries (LIB) have become the predominant energy storage systems being used in portable devices and electric vehicles [1], as they possess a high energy and power density [2]. Modern energy storage solutions require enhanced performance characteristics. Tailored battery electrodes, which are capable of interlocking on a micrometer scale have been observed to have a positive impact on cell characteristics at higher charging and discharging rates [3,4]. This effect even increases as the area loading of the electrode rises [5], as is required for high energy applications. There are only limited possibilities for fabricating three-dimensional geometries in conventional electrode production [6,7]. Slot die or doctor blade processing are typical techniques [2].

Novel approaches are being pursued with additive manufacturing (AM) technologies due to the unique properties, such as design freedom and multi-material processing [8,9]. In particular, droplet-based technologies yield a high potential for fabricating structures in the low micrometer range [9]. Droplet-based technologies include the different types of inkjet printing (IJP), common ones, such as drop-on-demand IJP, but also burgeoning ones, such as electrohydrodynamic jetting [10,11], as well as processes that are based on different physical principles, such as electrospinning [12]. However, these processes have only been qualified for a limited amount of materials yet [9]. At the same time, these droplet-based processes place challenging requirements on the dispersion to be printed. A homogeneous dispersion and high component stability must be ensured [13,14]. It is also necessary to adjust the properties of the dispersion to the nozzle geometry of the print head

to avoid nozzle clogging and ensure the formation and deposition of stable droplets [15]. The particle size distribution is one very important parameter of the dispersion, which must not exceed a certain limit value. This value depends on the nozzle diameter, which, in the case of common print heads, is generally around 50 μm [16]. Derby & Reis [17] defined a limit value for the particle diameter of 1/20 of the nozzle diameter. Hutchings et al. [18] specified a threshold value of 1/10. Guo et al. [13] stated that the particle diameter should be at least 20 to 50 times smaller than the nozzle diameter. This means that the particle size is restricted to a range of 1 μm to 5 μm.

The trend towards new manufacturing processes is accompanied by an increasing aqueous electrode processing, rather than processing based on N-Methyl-2-pyrrolidone (NMP). This leads to reduced costs and an improved environmental impact, since NMP has been found to be hazardous, teratogenic, and subchronic [19–21]. According to Zackrisson et al. [22], the transition from NMP to water results in a substantial reduction in $CO_2$ emissions throughout the life cycle. However, the low initial stability of the majority of active materials and conductive additives in an aqueous medium remains a challenge [23,24]. Organic electrode components, in particular, graphite and carbon black (CB), display low stability in water-based formulations. However, this effect is mitigated by the presence of carboxymethyl cellulose (CMC), which is the most common binder material used in aqueous dispersions. CMC acts as a thickener and, thus, enhances the stability of the dispersion components in water [25]. Nevertheless, stability can be further improved using additives. Indeed, this is indispensable to ensure good printability. Because the requirements placed on stability in conventional electrode processing are less stringent, the effect of additives on dispersion stability has not yet been subjected to thorough study.

Zhu et al. [26] investigated the stability of graphite in aqueous formulations with the aim of qualifying innovative heat transfer fluids. They achieved improved graphite dispersions by adding the non-ionic dispersant polyvinylpyrrolidone (PVP). The particle size distribution that they obtained was <1 μm for particle flakes with a diameter of 15 nm, although the PVP content was not indicated. Lee et al. [27] demonstrated the suitability of poly(acrylic acid) (PAA-NH) in increasing the dispersion of natural graphite in an aqueous medium in conventional LIB production. The scope of their investigation did not include an analysis of the particle size distribution.

González-García et al. [28] examined six different CB powders in aqueous solutions and achieved stabilization while using a non-ionic dispersant, Triton X-100 (TX-100). They investigated the adsorption on the basis of isotherms rather than the particle size distribution. Porcher et al. [29] examined the suitability of dispersants in stabilizing CB in an aqueous lithium iron phosphate (LFP) dispersion, focusing on dispersion time, equipment, and dispersant content. The dispersants investigated comprised a cationic type, hexadecyltrimethylammonium bromide (CTAB), a non-ionic type, sodium dodecyle sulphate (SDS), and TX-100. The most homogeneous distribution was achieved with TX-100, although the particle size distribution displayed a distinct fraction within the range of 1 μm and 10 μm.

Ueki et al. [30] examined the effect of PVP on graphite powders with a mean diameter of 100 nm as well as on CB powders with a mean diameter of 24 nm, respectively. Particle size distributions with large fractions above 1 μm were obtained with both materials.

While most of the previous studies concentrated on examining the impact of dispersants in conventional electrode dispersions or in applications outside the field of battery technology, this paper primarily aims at investigating their suitability in printing electrodes for LIBs. The focus is on graphite and CB materials that are suitable for LIB production. In addition to examining the particle size distribution, the application of dispersants is also evaluated with regard to the expected final cell properties. The paper concentrates on the dispersion behavior of organic electrode components, rather than ionic components, such as active materials in cathode dispersions. A systematic approach is developed to further generalize the findings of this study, which is based on a more profound detailing of the prevailing stabilizing mechanisms.

## 2. Theory of Stability, and Approach Pursued

Although both graphite and CB have a hydrophobic surface, they differ in their atomic structure and surface properties [31]. The purer the CB, the greater its properties, in particular its surface charge and non-polarity, approach those of graphite [31]. While graphite displays a homogeneous surface, that of CB is more heterogeneous. Depending on the degree of purity, it can contain a variety of different functional groups, such as carboxyl and hydroxyl groups, phenolic groups, and lactols [5,32]. Depending on the type of functional group, they display either an acidic or alkaline behavior in water [32].

Substances with non-polar surfaces can be stabilized by non-ionic surfactants, such as PVP and TX-100. The hydrophobic tail of the surfactant adsorbs at the non-polar surfaces, while the hydrophilic component extends into the liquid medium. For non-ionic surfactants, the stabilization mechanism is primarily based on electrosteric effects (a combination of electrostatic and steric stabilization).

The phenomena occurring in the diffusive ion layer dominate electrostatic stabilization. Equally charged ion layers repel each other, thus preventing the formation of aggregates and stabilizing the dispersion. The zeta potential is often used as an indicator of the magnitude of the electrostatic forces between colloidal particles and, thus, is determined by the surface charge density [33]. Consequently, the zeta potential is a function of the surface or particle charge, which is significantly influenced by the pH value [1]. According to Zhu et al. [26], the higher the absolute value of the zeta potential, the greater the electrostatic stabilization of a dispersion.

Steric stabilization is achieved when molecules adsorb at the particle surface and prevent the particles from approaching each other. A sufficient quantity of molecules must be adsorbed for the layer formed to be dense enough to shield the Van-der-Waals forces. Unlike electrostatic stabilization, steric stabilization is not reflected in zeta potential measurements [1].

Therefore, this paper investigates the following hypothese:

**Hypothesis 1.** *Graphite and CB nanoparticles can be stabilized in aqueous formulations, such that the dispersions meet the requirements for both printability and final cell performance.*

**Hypothesis 2.** *Even though CB is graphite-based, graphite and CB cannot be stabilized by the same non-ionic dispersants.*

The approach pursued in this paper is as follows. First, the cause–effect relationships between the indicators of instability in a dispersion and the measured quantities are analyzed. The predominant cause–effect relationships were experimentally investigated for graphite and CB dispersions containing either PVP or TX-100. Based on the findings, a systematic methodology is then derived, which aids in the utilization of dispersants in printable electrode dispersions for drop-on-demand IJP processes.

## 3. Materials and Methods

### 3.1. Synthesis of Dispersions

The particle-water mixtures were prepared in accordance with the flow chart in Figure 1. All of the mixing sequences were conducted at a temperature of 20 °C and ambient pressure.

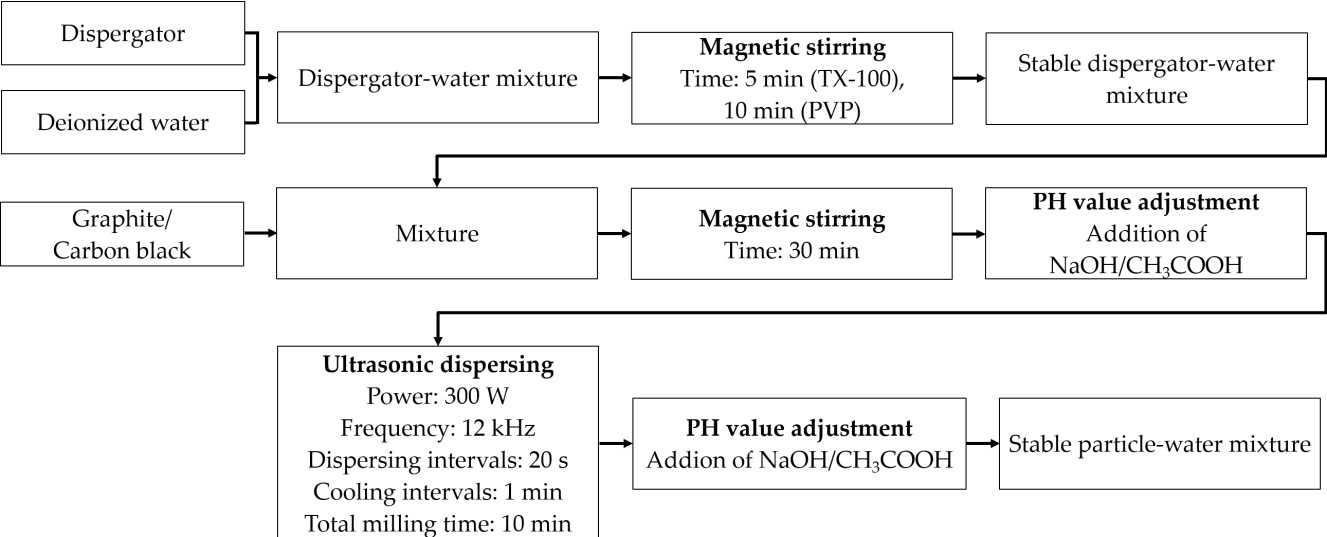

**Figure 1.** Flow chart showing the preparation process of the particle-water mixtures.

First, the dispersant—either TX-100 (Sigma-Aldrich, Continental, USA) or polyvinylpyrrolidone (PVP) (K17, BASF, Germany)—was stirred into deionized water. Second, either high-purity graphite nanoparticles (Nanografi, Turkey) that were suitable for LIB production with a mean diameter of 30 nm, or CB nanoparticles (C65, Timcal, Switzerland) with a mean diameter of <50 nm were added to the as-prepared mixture and stirred for 30 min. Subsequently, the pH value was adjusted by adding either sodium hydroxide solution (NaOH) or acetic acid (CH$_3$COOH), as required. The mixture was then dispersed using an ultrasonic homogenizer (FS-300N, Vevor, China). The particles are released from agglomerates, causing an increase in the total free particle surface. This leads to a change in the pH value, necessitating a further pH value adjustment to reach the target value.

### 3.2. Analysis of Dispersion Properties

#### 3.2.1. Measured Quantities

The presence of agglomerates and the occurrence of sedimentation are indicators of an unstable nanoparticle-based dispersion. Agglomerates can amplify the process of sedimentation, since larger and heavier particles settle faster. Both of the indicators can be quantified by various measurands (see Table 1). These quantities primarily comprise viscosity, zeta potential, particle size distribution, and a quantity resulting from an optical evaluation, such as the average gray value.

Viscosity

Assuming that they are of the same composition and preparation, a well-dispersed sample has a lower viscosity than a poorly dispersed one that contains agglomerates. Agglomerations occur when the forces acting between the particles—in particular, the Brownian, hydrodynamic, and colloidal forces—are not balanced, and attractive forces predominate in the dispersion [34]. These agglomerations form a three-dimensional network, and the dispersion becomes gel-like and it has a higher viscosity [34].

A high standard deviation is another indication of the presence of agglomerates, as they have a strong effect on the reproducibility of the viscosity values. The measuring gap must be significantly larger than the biggest particles in the dispersions to determine viscosity using a rheometer [35]. However, agglomerations exhibit diameters many times larger than single particles. As a result, the measuring gap during the measurement may not be sufficiently large, in which case the viscosity of the agglomerate is measured instead of that of the dispersion [36]. Because of the rotation of the rheometer, this can occur with varying frequency and intensity, which results in large standard deviations [36].

A decrease in viscosity implies the occurrence of sedimentation. The more the particles settle, the smaller their influence on the viscosity. According to the equation that was formulated by Krieger–Dougherty and Einstein, the viscosity of a dispersion increases as the volume fraction of the particles in the liquid phase increases [37]. However, sedimentation causes particles to settle at the bottom of the vessel, which reduces the volume fraction of particles in the remainder of the dispersion [37]. The viscosity then approaches that of the pure fluid, which is lower [37].

Zeta Potential

Low absolute zeta potential values are an indication of the presence of agglomerates [1,38,39]. The repulsion forces between the particles are not sufficient for preventing them from approaching each other [1,39]. It has to be considered that small changes in the pH value can already have a high impact on zeta potential values [30].

Particle Size Distribution

The particle size distribution is a reflection of the particle dispersal. If particles are detected that are larger than the initial particles, it is a an indication that agglomerates have formed.

Optical Evaluation

An optical evaluation of the particles (if their size is within the visible range) enables their characterization in terms of sedimentation and agglomeration. Isolated particles represent agglomerates. The formation of a sediment and a decrease in the contrast of a dispersion indicate sedimentation.

The first three quantities that are listed above (viscosity, zeta potential, and particle size distribution) are preferable to an optical evaluation, as they are easier to determine and more reliable. Based on Table 1, the following cause–effect relationships were found to be relevant to a comprehensive evaluation of a dispersant:

- Influence of the dispersant content on viscosity.
- Effect of the pH value on zeta potential.
- Impact of the pH value on viscosity.
- Determination of the particle size distribution.

**Table 1.** Quantities determining the stability of dispersions.

| Indicators / Variables | Agglomerates | Sedimentation | Reported by |
|---|---|---|---|
| Viscosity | High viscosity values (same composition and preparation), high standard deviation | Decreasing viscosity at a constant shear rate | [34,36,37] |
| Zeta potential | Low absolute values | - | [1,38,39] |
| Particle size distribution | Higher than particle size of single nanoparticles | - | [26,29] |
| Gray value from optical evaluation | Isolated particles | Formation of a sediment, decreasing contrast of the dispersion | [40] |

Viscosity, zeta potential, and particle size distribution are sensitive to sedimentation. In the case of pronounced sedimentation, an optical evaluation can provide an initial assessment of the dispersion.

### 3.2.2. Measuring Equipment and Methods

Rheological properties were measured with a rotational rheometer (Kinexus lab+, Netzsch, Germany). A 40 mm plate–plate geometry with a sample gap of 0.1 mm was employed in conjunction with a passive solvent trap. Each sample was equilibriated

at 25 °C prior to the measurement. A single viscosity measurement was conducted at 80 1/s for 5 min, capturing 61 single point measurements. Zeta-potential measurements were conducted with a zeta potential analyzer (Zetasizer Nano ZSP, Malvern Panalytical, Malvern, UK). To enable this, the dispersions were diluted in deionized water at a ratio of 1:250. The pH value was determined with a pH measuring instrument (PH-100 ATC, Voltcraft, Switzerland) with an absolute accuracy of 0.2 pH. The particle size distribution was determined while using a static laser diffraction analyzer (SALD-2201, Shimadzu, Japan). The as-prepared dispersions were diluted in deionized water at a ratio of 1:500. The prepared dispersions were poured into transparent test vessels to enable optical assessment. Camera images (D60, Canon, Japan) of the samples were taken at regular intervals. The images were inverted using GIMP image processing software. The average gray value was calculated from a histrogram analysis. The minimum gray value was 0 and the maximum average gray value 255, with 0 being assigned to black and 255 to white. A low average gray value thus denotes that the dispersion is comparably dark in color. This occurs when the particles are homogeneously distributed in the solution and they have not yet settled down to the bottom of the vessel.

## 4. Results and Discussion

### 4.1. Stabilization of Graphite

#### 4.1.1. Influence of Dispersant Content on Viscosity

The rheological behavior of the graphite dispersions was investigated for different PVP and TX-100 contents, respectively. The rheological behavior of a pure PVP and a pure TX-100 solution were also examined. The pH value of the PVP samples was adjusted to 9.0–9.5, since Zhu et al. [26] postulated the pH value at the lowest zeta potential to be 9.0, while Ueki et al. [30] put it at 9.5. For TX-100, no suitable data regarding pH were found in the literature. Therefore, the pH value of the TX-100 samples was adjusted to the same pH value range as the PVP samples. The viscosity curves are given in Figures 2 and 3. Graphite content refers to the total mass consisting of water and graphite. Dispersant content relates to the graphite content. Unless stated otherwise, this also applies to all further measurements. The standard deviation is based on 61 single point measurements.

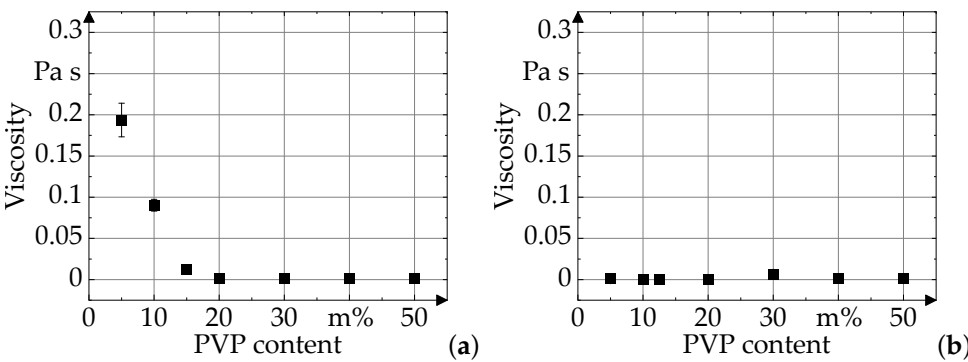

**Figure 2.** Viscosity as a function of polyvinylpyrrolidone (PVP) content for (**a**) an aqueous dispersion containing 10 m% graphite nanoparticles (see Table A1) and (**b**) a PVP solution without nanoparticles (see Table A2).

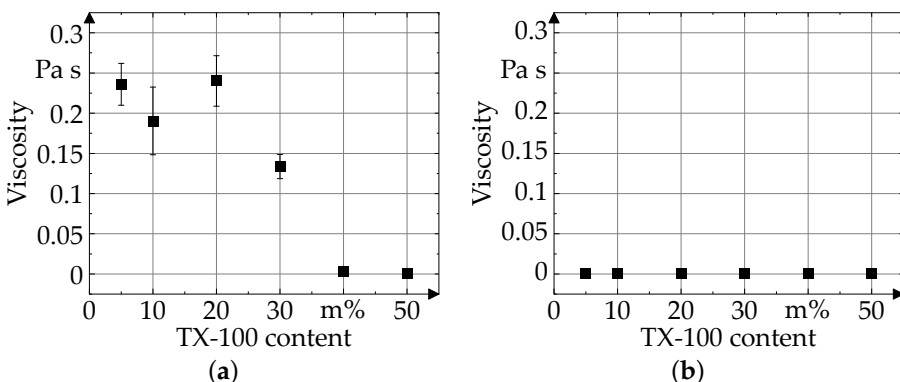

**Figure 3.** Viscosity as a function of TX-100 content for (**a**) an aqueous dispersion containing 10 m% graphite nanoparticles (see Table A3) and (**b**) a TX-100 solution without nanoparticles (see Table A4).

The viscosity curves for both graphite dispersions (PVP and TX-100) show a similar progression. Below, a PVP content of 20 m% or a TX-100 content of 40 m%, respectively, the viscosity of the graphite dispersion gradually improves as the dispersant increases. In this range, the graphite nanoparticles are successively coated by PVP or TX-100 molecules, respectively, in turn weakening the interparticle attraction forces [39]. A PVP content of 20 m% or a TX-100 content of 40 m% indicates the onset of a plateau in viscosity for both pH values. At this point, the added PVP or TX-100 molecules are fully adsorbed at the surface of the graphite particles, marking the highest stability of the dispersion. The viscosity values in the range of the plateau correspond well with those of pure PVP or TX-100 in an aqueous medium (see Figures 2b and 3b). A further increase in PVP or TX-100 content does not lead to any significant change in viscosity and thus enhanced stability. Because the graphite surface is fully coated, the excess PVP or TX-100 molecules remain in the water. Hence, the viscosity of the graphite dispersion is gradually determined by the viscosity of the PVP or TX-100.

The observations in the viscosity curves are also reflected in the appearance and behavior of the dispersions. The PVP dispersions seemed to be well dispersed with no visible agglomerates. The higher the PVP content, the more homogeneous the dispersion. However, the TX-100 dispersions displayed a number of isolated agglomerates. Furthermore, a foamy behavior of the dispersions was observed as the TX-100 content increased. The differences in the external appearance of the dispersions can be attributed to the formation of micelles. Micelles are formed when the critical micelle concentration of a surfactant in a solution is reached. The hydrophilic parts of the surfactant molecules then align themselves with the adjacent water molecules, whereas the hydrophobic parts cluster together, forming a separate phase [41]. Varying statements can be found regarding the critical micelle concentration of the surfactant TX-10, whereby Patist et al. [41] determined that the value highly depends on the measuring principle. Sigma Aldrich [42] specifies a range of 0.22 and 0.24 mmol/L, which is equivalent to about $3.95 \times 10^{-3}$ m% in the investigated dispersion system. This illustrates that micelles are present, even at marginal TX-100 concentration. In contrast to low molecular weight surfactants, such as TX-100, amphiphilic side-chain polymers, such as PVP, do not exhibit a critical micelle formation [43].

### 4.1.2. Effect of pH Value on Zeta Potential

The zeta potential was measured for the concentrations at which a plateau was reached in the viscosity curves, corresponding to 20 m% for PVP and 40 m% for TX-100 (see Figures 2 and 3). Figure 4 shows the influence of the pH value on the zeta potential. The standard deviation was taken from three samples on the basis of three single measurements. The pH value was adjusted by a dosing needle with an internal diameter of 0.61 mm. The measured values do not display regular intervals because the addition of such small quantitites is subject to a certain degree of inaccuracy.

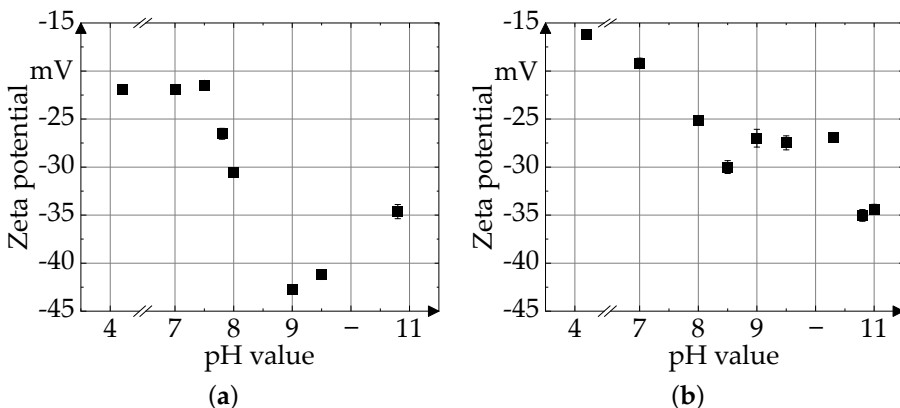

**Figure 4.** Zeta potential of graphite dispersions at various pH values for (**a**) a PVP content of 20 m% (see Table A5) and (**b**) a TX-100 content of 40 m% (see Table A6).

The pH value dependence was primarily investigated around the neutral or in the slightly basic range, as previous works had shown that anodic electrode dispersions with a pH value in this range positively influenced the final cell performance [44]. In the acidic range (pH = 4.2), the zeta potential is −22 mV for the PVP dispersion and −16 mV for the TX-100 dispersion. A strong decrease in the zeta potential can be observed with increasing pH for both dispersion types. The electrostatic repulsion forces among the particles increase successively, which results in higher stability of the dispersions [38]. For the PVP dispersion, a minimum zeta potential of approximately −45 mV is attained at a pH value of 9. This was found to correlate well with previous studies. Ueki et al. [30] stated that the most suitable pH value with regard to stability is approximately 9, while Zhu et al. [26] specified for this a pH value of 9.5. The zeta potential curve of the TX-100 dispersion reaches its minimum at a pH value of 10.75. This is at −34 mV. The minimum zeta potential indicates that the most stable dispersion is obtained at this pH level. A further increase in the pH value leads to an increase in the zeta potential. This is due to the increased ion concentration in the aqueous medium, according to Zhu et al. [26]. This has a compressive effect on the diffuse double layer, leading to a reduced zeta potential.

Varying statements regarding the zeta potential and its dependence on stability can be found in the literature. Van Nguyen et al. [39] and Greenwood [1] determined that dispersions with a zeta potential of less than −30 mV and more than +30 mV are stable. According to this definition, the graphite dispersion with PVP is stable within a pH value range between 8 and 11. The graphite dispersion in the case of TX-100 only appears to be stable for pH values that are between 10.7 and 11. However, White et al. [33] stated that even zeta potential values of less than −15 mV or more than 15 mV can indicate sufficient stabilization. According to this limitation, the graphite dispersions with PVP and TX-100 are stable throughout the entire investigated pH value range of 4 to 11. In this case, no pH value adjustment is required, since the natural pH value of the investigated dispersion is around 4. It is worth mentioning that all of the dispersions with TX-100 displayed isolated agglomerates, characterizing a not-stable state. This makes it clear that a strict classification of dispersions into stable and non-stable according to the given zeta potential limits is not applicable for all dispersion types. It allows for an evaluation of dispersions with regard to electrostatic stabilization, but not to steric stabilization [1]. However, it was observed that reproducible values can only be obtained for electrostatic stabilization if no strong sedimentation occurs.

It also appeared that adjusting the pH value in the basic range of dispersions containing TX-100 required the addition of more NaOH than for dispersions with PVP. To gain a deeper understanding of the influencing factors, the correlations between pH value and both dispersant and particle content were investigated. Figure 5 illustrates the pH dependence of PVP without and with graphite nanoparticles. Figure 5a displays the pH value for various PVP contents without particles, while Figure 5b shows the pH values of disper-

sions with a different graphite content and at a constant PVP fraction of 20 m%. Figure 6 illustrates the same relations for TX-100. The influence of the graphite nanoparticles on the pH value could not be individually examined, as graphite displays low initial stability in an aqueous medium. Thus, it was not possible to create a homogeneous dispersion. In the samples without particles, the amount of added PVP or TX-100 content related to the amount of 10 m% particles. The standard deviation was calculated from three single measurements.

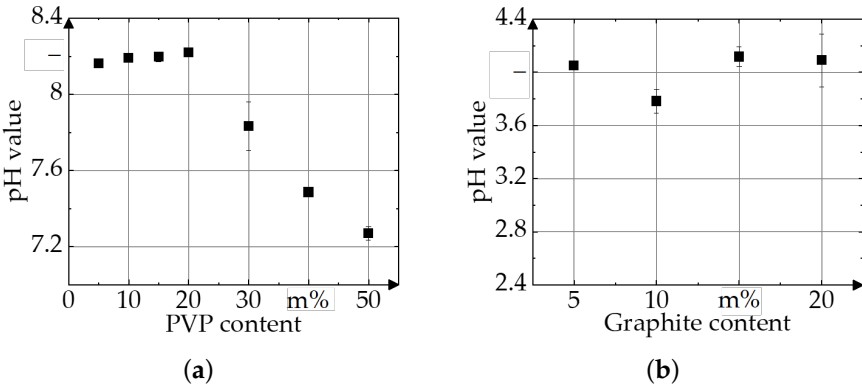

**Figure 5.** PH value as a function of (**a**) PVP content for a solution without nanoparticles (see Table A7) and (**b**) graphite content for aqueous dispersions containing 20 m% PVP (see Table A8).

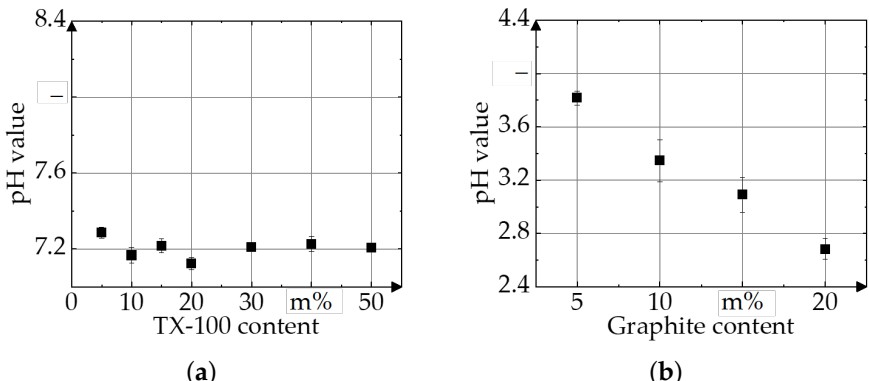

**Figure 6.** PH value as a function of (**a**) TX-100 content for a solution without nanoparticles (see Table A9) and (**b**) graphite content for aqueous dispersions containing 40 m% TX-100 (see Table A10).

Figure 5a shows that, with a PVP fraction of between 5 m% and 20 m%, the pH value is approximately constant. In the range between 20 m% and 50 m%, the pH value decreases roughly linearly from 8.2 to 7.3. For the dispersion containing a constant amount of 20 % PVP and a varying content of graphite, no distinct pH value dependence can be noted between a graphite fraction of 5 m% and 20 m% (see Figure 5b).

As is evident from Figure 6a, the pH value as a function of TX-100 content fluctuates by about 0.2 measurement units. This deviation lies within the limits of accuracy of the measuring instruments. Thus, it appears that the TX-100 content has no significant influence on the pH value in the investigated value range. However, the dispersion with a constant content of 40 m% TX-100 and a varying amount of graphite shows a pH value dependence (see Figure 6b). The pH value decreases almost linearly from 3.8 at a proportion of 5 m% graphite to 2.7 at a proportion of 20 m%. Comparing Figures 5b and 6b shows that, for TX-100, the pH value shifts more towards acidity than with PVP. While the differences were moderate for a solid content of 10 m%, they were substantial for higher solid contents. This corresponds with the fact that the required amount of NaOH is higher with TX-100 than with PVP at a solid content of 10 m%.

Without graphite particles, there are only interactions between the water and existing dispersant.

Figure 5a corroborates the occurrence of a hydrolysis reaction. Water releases a hydrogen atom, which causes a transformation of the lactam ring of the PVP to an amino acid group. The amino acids, in turn, react with the water, resulting in the production of $H_3O^+$ ions, which directly determine the pH value. For small amounts of PVP, the ions do not yet have a considerable effect on the pH value of the total dispersion. It appears that, between 20 and 30 m%, a threshold value is exceeded, from which the ions directly influence the pH value of the total system. The observations in Figure 6a correspond well with the findings of Fernández-Merino et al. [45]. They stated that TX-100 does not show a pH value dependence due to the non-existence of functional groups that can react with water.

When adding graphite particles to the water-dispersant mixture, the dispersants may interact with both the particles and water. Because of the non-existence of functional groups at the graphite surface, PVP adsorbs homogeneously, forming a uniform film [30]. The hydrophilic tail group adsorbs at the hydrophobic graphite surface, while the hydrophilic polar head group is oriented towards the water [31]. It is assumed that the lactam group is then less accessible to the water molecules, which makes a hydrolysis reaction more difficult. González-García et al. [28] stated that an equilibrium exists for TX-100 between the free dispersant in the solution and the adsorbed dispersant at the particle surface. TX-100 adsorbs at the graphite particles more irregularly than PVP, so the graphite surface is not fully coated with TX-100. This may allow for interactions between the graphite and water. These interactions lead to exchanges of hydrogen atoms, which, in turn, lead to a considerable pH value dependence (see Figure 6b).

### 4.1.3. Impact of pH Value on Viscosity

The rheological behavior of the graphite dispersions as a function of dispersant content was investigated for two different pH value ranges (see Figures 7 and 8). For PVP, the pH value that is determined in the zeta potential measurements lies within the pH value range identified from the literature. The measurements were repeated at a lower pH value, more towards the neutral range. For TX-100, the assumed pH value range, based on PVP, does not correspond with the one that was determined from the zeta potential measurements. Therefore, the second analysis was performed in the pH value range that resulted from the zeta potential evaluation. The standard deviation was based on 61 single point measurements.

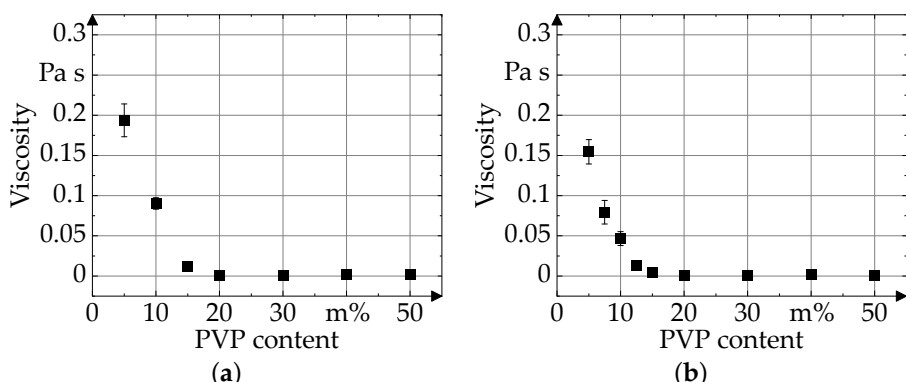

**Figure 7.** Viscosity as a function of PVP content for aqueous dispersions containing 10 m% graphite nanoparticles at varying pH values (**a**) pH = 9.0–9.5 (see Table A1) and (**b**) pH = 7.8 (see Table A11).

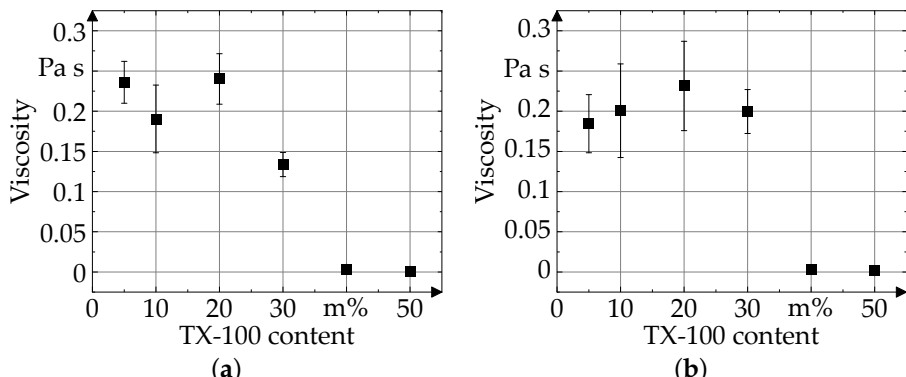

**Figure 8.** Viscosity as a function of TX-100 content for aqueous dispersions containing 10 m% graphite nanoparticles at varying pH values (**a**) pH = 9 (see Table A3) and (**b**) pH = 10.75–11 (see Table A12).

It can be seen that the progression of the viscosity displays similar pH values with both PVP and TX-100, respectively.

There are small differences in the viscosity values for a low dispersant content. With a small amount of PVP, the viscosity values are slightly lower for a pH value of 7.8 than for a pH value of 9.0, which resulted from the zeta potential measurements. This can be explained in terms of the Péclet number, which represents the relationship between convective and diffusive transport in dispersions [46]. A Brownian dispersion is present with small Péclet numbers, which are primarily determined by small particles, according to this characteristic value. It is likely that the particles no longer sedimented due to gravity, with a reversible equilibrium structure being able to establish itself [36]. Small particles floating in the aqueous medium resulted in increased viscosity. At a low TX-100 content, the viscosity values for both pH values were subject to a high standard deviation. For this reason, no definite trend can be identified. However, in the plateau area, the viscosity values were almost identical for both dispersants and pH values. This provides strong evidence that the pH value does not have a significant impact on the stabilized dispersions.

### 4.1.4. Determination of Particle size Distribution

The particle size distribution was analyzed for the graphite dispersions at a pH value of 9.0–9.5 for PVP and at a pH value of 10.75–11 for TX-100 (see Figure 4). The dispersion with TX-100 contained the minimum suitable dispersant content (see Figure 9a). The dispersion with PVP was even investigated for a dispersant content of 15 m%, which is slightly lower than the minimum dispersant content (see Figure 9b). The standard deviation resulted from three single measurements.

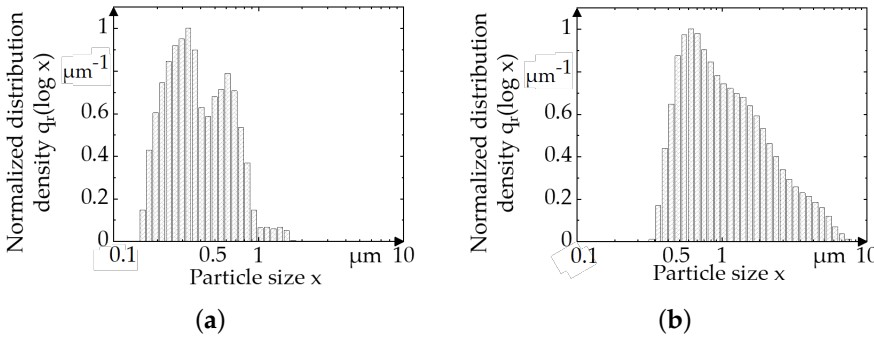

**Figure 9.** Particle size distribution of the aqueous dispersions containing 10 m% graphite nanoparticles and (**a**) 15 m% PVP (see Table A13) and (**b**) 40 m% TX-100 (see Table A14).

It can be seen that the graphite particles mixed with PVP are predominantly dispersed as nanometeric particles, although the dispersant content is slightly lower than that iden-

tified in the rheological characterization. The obtained particle size distribution is not larger than the required threshold range between 1 μm and 5 μm. The majority of particle fractions are even below 1 μm. However, the graphite dispersion dispersed with TX-100 shows a fraction of giant agglomerates of the order of several microns. Thus, these results confirm that PVP is more effective than TX-100 in dispersing graphite nanoparticles in an aqueous medium. The resulting particle size distribution meets the requirements of the inkjet printing process.

### 4.1.5. Impact of Dispersant on Final Cell Performance

The qualification of electrode dispersions is subject to a conflict of objectives. On the one hand, the electrode dispersions are to be adapted to the inkjet printing process. The aim is to fully exploit the advantages that result from the change in production process and the associated design freedom, such as design integration and possibilities for individualization [47]. On the other hand, the electrical competitiveness compared to conventional state-of-the-art compounds has to be ensured. Therefore, the amount of inactive materials, such as additives, in an electrode dispersion has to be as low as possible. This also applies to the addition of acids or alkalis, even if the amount is marginal. The results showed that, for the examined graphite dispersions, no addition is required, as the dispersions were stable throughout the entire investigated pH value range. This demonstrates that the stability of the analyzed graphite dispersions with PVP is not highly influenced by the pH value.

PVP has a hydrophilic and hydrophobic tail group. Thus, it exhibits both polymer and dispersant properties. Because of these characteristic properties, PVP can also inherit the function of the binder, which is indispensable for adhesion and cohesion in an electrode. In various studies, binder materials that are based on PVP have already been successfully used in conventional electrode production [48–50]. In addition, it is expected that the PVP content can be further reduced by using spheric instead of flaky nanoparticles due to the significantly smaller specific surface area (BET).

### 4.2. Stabilization of CB

The test procedure that was carried out for graphite dispersions was intended to also have been performed for CB. However, strong sedimentation occurred during the preparation of the dispersions for rheological and zeta potential measurement with both PVP and TX-100, and it was not possible to prepare a dispersion that would have been stable for the duration of the measurements. The measurements showed non-reproducible results with very high standard deviations. Therefore, an optical evaluation was chosen for CB to obtain an initial assessment regarding the suitability of PVP and TX-100.

The sedimentation behavior was studied for aqueous CB dispersions containing varying contents of PVP and TX-100. Particle settling and sedimentation was observed with all investigated dispersions. In order to characterize the intensity of the sedimentation, the average gray value was plotted for the dispersion as-prepared, after 1 h, and 12 h, with both PVP (see Figure 10) and TX-100 (see Figure 11).

A clear difference in the average gray values can be observed between the dispersions with and without dispersant for both PVP and TX-100. The average gray value of the dispersions without dispersant are significantly higher than those for the dispersions with PVP or TX-100. The minimum average gray value for the as-prepared dispersion is at 30 m% PVP and TX-100, respectively.

The average gray values that were obtained for the dispersions with TX-100 are considerably lower than for the dispersions with PVP. However, it is evident that the change in the gray value over time is more significant for TX-100 than for PVP, although the values are still lower for TX-100 than for PVP after 12 h.

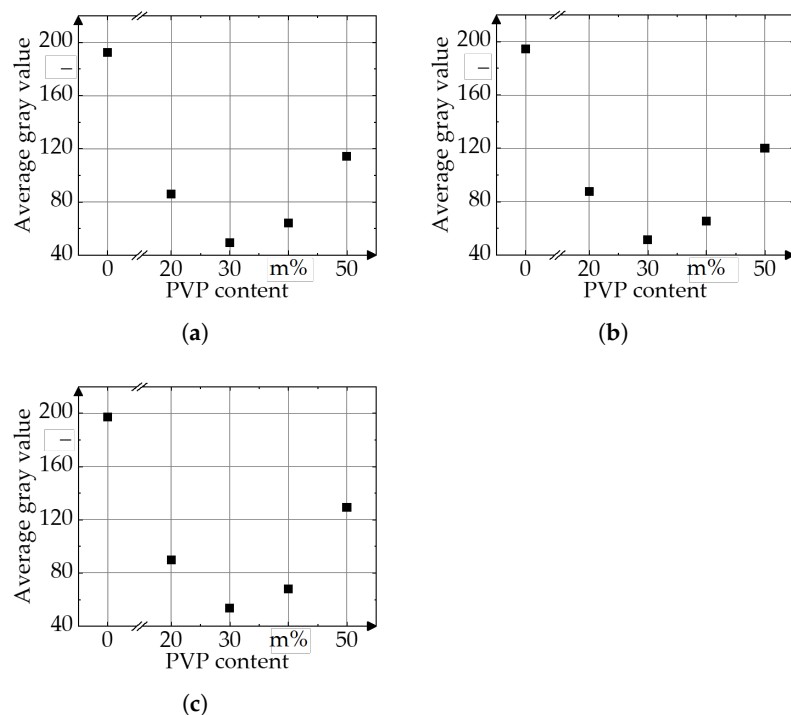

**Figure 10.** Average gray value of carbon black (CB) dispersions as a function of PVP content at different times (**a**) as-prepared (**b**) after 1 h and (**c**) after 12 h (see Table A15).

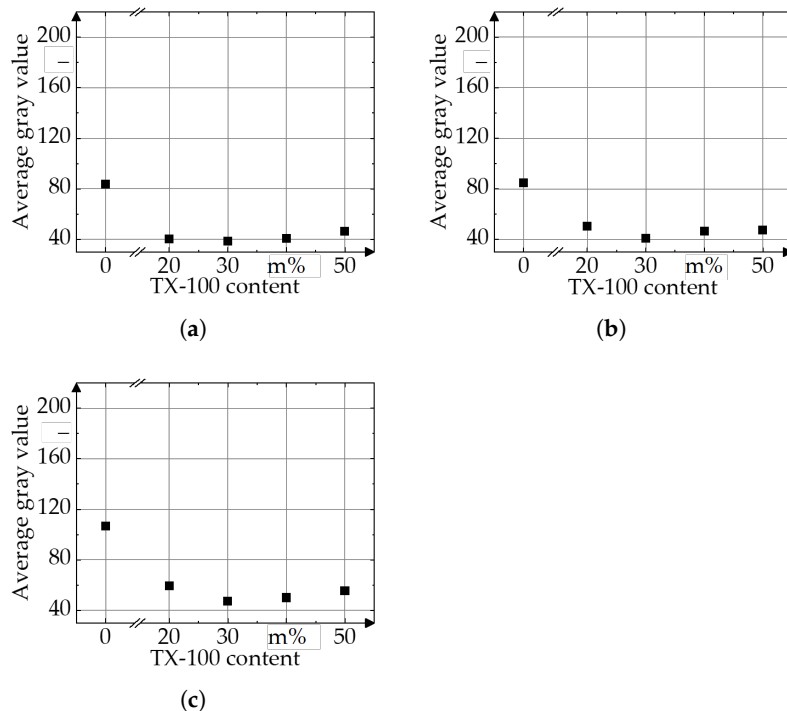

**Figure 11.** Average gray value of CB dispersions as a function of TX-100 content at different times (**a**) as-prepared (**b**) after 1 h and (**c**) after 12 h (see Table A16).

These observations make it clear that TX-100 decelerates sedimentation more effectively than PVP, but it is still not enough to obtain a stable dispersion. It is assumed that the addition of TX-100 breaks up the largest particle aggregates, the agglomerates, while the smaller aggregates remain. Agglomerates and aggregates differ in their forces of

attraction. Agglomerates are formed by the weak attractive forces between aggregates [51]. Aggregates are formed by strong attractive forces between the individual particles [51]. According to Bauer et al. [52], aggregates should be preserved, even if this results in larger particles. These aggregates have a strong impact on cluster formation in an electrode. The resulting clusters consist of binding agents with enclosed or attached CB particles. The embedded CB particles are needed in order to build up a conductive matrix that covers the entire cluster volume. Aggregated CB particles are more suitable than primary CB, since the former are stacked in a denser configuration [52]. A binder material, such as carboxymethyl cellulose (CMC), is added in the course of preparing an electrode dispersion. It is assumed that the resulting thickening additionally inhibits the sedimentation of smaller aggregates [25].

Particle size was subsequently investigated while using syringe filters (Micropur GF 1 µm, Altmann Analytik, Germany). Virtually any residues remained in the mesh. This corroborates the vast majority of CB aggregates being of smaller size than the threshold $x_t$ for inkjet printing.

### 4.2.1. Methodology

Based on the empirical results, a methodology was derived to support the selection and application of a suitable dispersant for organic electrode components (see Figure 12).

Initally, dispersions of varying dispersant content are prepared and their sedimentation behavior is evaluated.

If no strong sedimentation is observed, the viscosity of the dispersions $\eta$ is determined as a function of the dispersant content $\omega$. Measurements are performed at a suitable pH value (range) $pH_1$, as stated in the literature. The minimum suitable dispersant content $\omega_{min,1}$ is present when the viscosity reaches a minimum. The zeta potential $\zeta$ is measured as a function of the pH value. The investigated dispersions contain the minimum amount of dispersant $\omega_{min,1}$ identified in the previous viscosity measurements. This enables both the pH value dependence of the zeta potential and the pH value at the lowest zeta potential $pH_2$ to be more precisely determined. Based on this, the viscosity measurements are repeated for the pH value, for which the zeta potential marked the highest absolute value $pH_2$. This allows for evaluating how the minimum dispersant content changes with the pH value. The particle size distribution of the dispersions is then measured. The dispersions contain the minimum amount of dispersant $\omega_{min,2}$ and they are adjusted to the pH value $pH_2$. The highest detected particle size represents the threshold particle size $x_t$.

In the event of sedimentation, a different procedure has to be pursued. Optical evaluation is then performed by examining the average gray value $GW$ as a function of the dispersant content. The minimum dispersant content $\omega_{min}$ is that of the dispersion exhibiting the lowest average gray value. To identify a suitable pH value $pH_2$, the test series is repeated for different pH values at a constant dispersant content $\omega_{min,1}$. The average gray value is then measured for the pH value for which the average gray value marked the lowest value $pH_2$. Based on this, the particle size can be analyzed while using a filtering method. The pore size must be selected, such that it represents the upper limit of the particle size $x_t$. For an initial assessment, it is sufficient to analyze the gray value as a function of the dispersant content at $pH_1$ as well as the filtering method. Thus, the subsequent steps are omitted in the initial analysis. Because of the expected higher basic stability resulting from the addition of binder in the next step, it is expected that a full characterization is feasible on the basis of quantifiable variables comprising viscosity, zeta potential, and particle size distribution.

To assess printability in terms of particle size, $x_t$ can be compared with the threshold value range that is stated in the rules of thumb in the literature. Based on this, the dispersions are categorized into printable and non-printable. Finally, the expected impact of the dispersants on the final cell properties is evaluated.

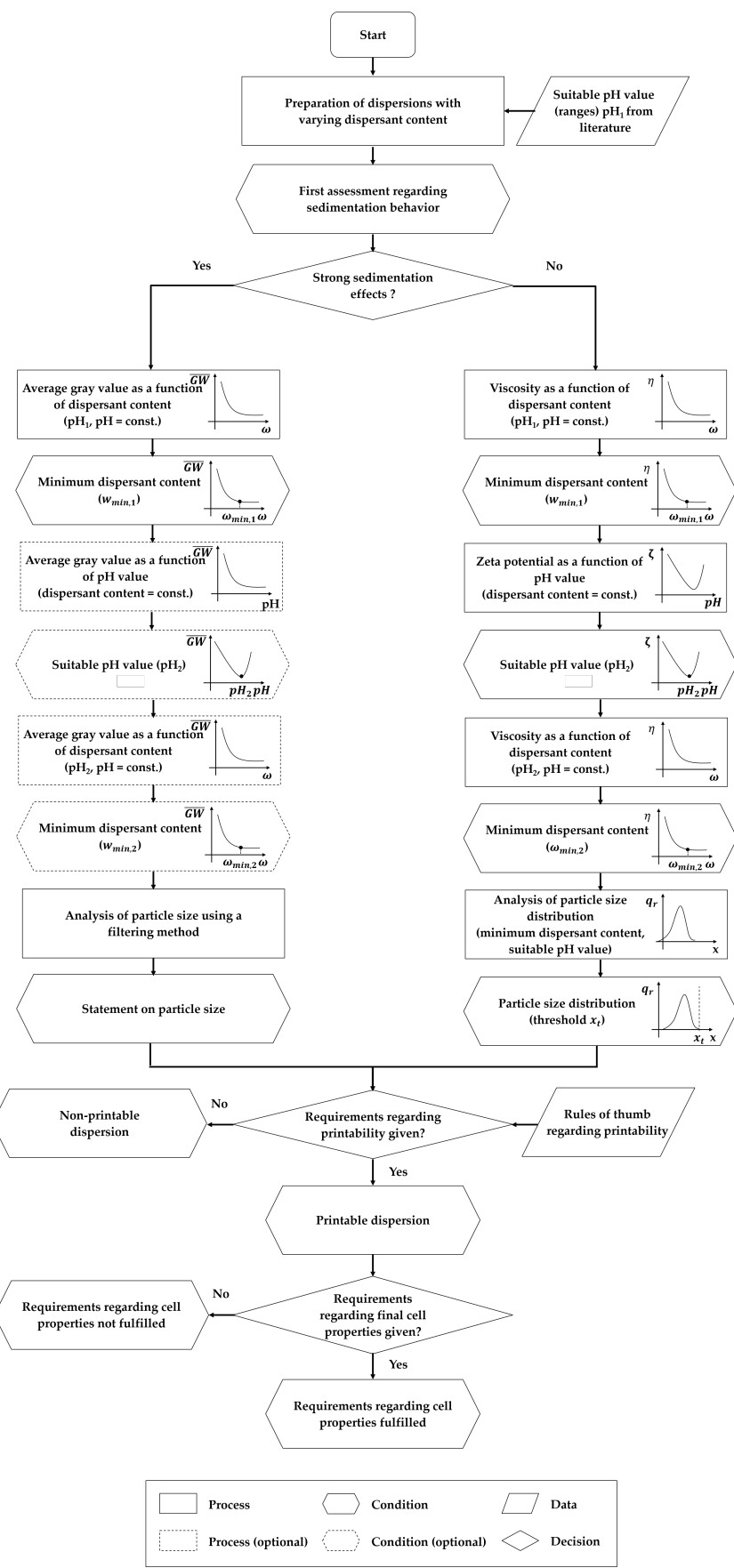

**Figure 12.** Flow chart showing the methodology; the symbols of the illustration were selected according to DIN 66001 [53]).

## 5. Conclusions

Organic electrode components, especially graphite and carbon black, display low inherent stability in aqueous formulations. Good dispersing and, in turn, stabilization behavior is particularly important when it comes to qualifying electrode dispersions for inkjet printing processes. Additives are capable of further improving stability. An empirical methodology was presented, which aids the application of suitable dispersants with organic electrode components. Empirical investigations revealed appropriate dispersants and operating parameters for use in inkjet printing processes. It is likely that the systematic approach can be applied to graphite and carbon black materials with slightly different properties. The findings of this paper can be summarized, as follows:

- Polyvinylpyrrolidone (PVP) was found to be a suitable dispersant for the stabilization of graphite. The particle size distribution that is achieved with PVP is below the threshold required for processing in inkjet printing. For the given graphite nanoparticles with an average particle size of 50 nm, a PVP content of 20 m% with respect to particle quantity was identified as appropriate.
- Triton X-100 favoured an enhanced dispersibility of carbon black. It appears that TX-100 is capable of dissolving secondary agglomerates, although primary aggregates are still present in the dispersion. From an electrochemical point of view, primary aggregates should not be destroyed. Filter examinations revealed that the size of the majority of aggregates is lower than the limit value for inkjet printing. It is assumed that the addition of a binder material, such as carboxymethyl cellulose (CMC), additionally inhibits the sedimentation of the smaller primary aggregates by virtue of its thickening effect.

In conclusion, the findings corroborated both of the hypotheses. On the one hand, the dispersal of graphite and carbon black nanoparticles can be ensured using additives that keep the particle size distribution below the limit for inkjet printing. On the other hand, graphite and carbon black cannot be stabilized by the same non-ionic surfactants, despite their similar chemical origin.

However, preparing a stable dispersion from organic components is only one of the first steps in qualifying an electrode dispersion for inkjet printing processes. In this context, further investigations are necessary to determine as to whether the addition of a binder, such as CMC, is sufficient for stabilizing primary carbon black aggregates. The next step is to achieve a stable dispersion from the entirety of the components of both anode and cathode dispersions. In addition to the organic components, these include the active cathodic materials and the binder, in the mass fractions relevant to the later electrode. It remains to be examined whether PVP can also act as a binder in the electrode system and ensure the required cohesion and adhesion. Further investigations must be carried out with regard to the suitability and the optimum content.

**Author Contributions:** Conceptualization, C.G.K.; methodology, C.G.K.; performing experiments and analyzing data, C.G.K. and M.L.; writing–original draft preparation, C.G.K.; writing–review and editing, M.L., J.-L.L., A.B. and M.F.Z.; visualization, C.G.K.; supervision, M.F.Z., A.B.; funding acquisition, C.G.K., A.B. and M.F.Z. All authors have read and agreed to the published version of the manuscript.

**Funding:** The authors express their sincere gratitude to the Federal Ministry of Education and Research (BMBF) for financially supporting the research on the presented topic within the research cluster ExZellTUM III (grant Number 03XPO255).

**Data Availability Statement:** Not applicable.

**Conflicts of Interest:** The authors declare no conflict of interest.

## Appendix A

**Table A1.** Viscosity values as a function of PVP content for an aqueous dispersion containing 10 m% graphite nanoparticles at a pH value of 9.0.

| Dispergator Content in m% | Viscosity in Pa s | Standard Deviation in Pa s |
|---|---|---|
| 5 | $1.94 \times 10^{-1}$ | $0.20 \times 10^{-1}$ |
| 10 | $0.90 \times 10^{-1}$ | $7.19 \times 10^{-3}$ |
| 15 | $1.25 \times 10^{-2}$ | $2.09 \times 10^{-3}$ |
| 20 | $1.41 \times 10^{-3}$ | $1.03 \times 10^{-4}$ |
| 30 | $1.31 \times 10^{-3}$ | $1.03 \times 10^{-4}$ |
| 40 | $1.59 \times 10^{-3}$ | $2.59 \times 10^{-4}$ |
| 50 | $1.65 \times 10^{-3}$ | $1.83 \times 10^{-4}$ |

**Table A2.** Viscosity values as a function of PVP content for a PVP solution without nanoparticles.

| Dispergator Content in m% | Viscosity in Pa s | Standard deviation in Pa s |
|---|---|---|
| 5 | $9.18 \times 10^{-4}$ | $5.23 \times 10^{-5}$ |
| 10 | $6.11 \times 10^{-4}$ | $1.35 \times 10^{-5}$ |
| 12.5 | $6.62 \times 10^{-4}$ | $1.20 \times 10^{-5}$ |
| 20 | $7.79 \times 10^{-4}$ | $4.34 \times 10^{-5}$ |
| 30 | $6.68 \times 10^{-3}$ | $1.32 \times 10^{-3}$ |
| 40 | $1.42 \times 10^{-3}$ | $1.05 \times 10^{-4}$ |
| 50 | $1.50 \times 10^{-3}$ | $1.06 \times 10^{-4}$ |

**Table A3.** Viscosity values as a function of TX-100 content for an aqueous dispersion containing 10 m% graphite nanoparticles at a pH value of 9–9.5.

| Dispergator Content in m% | Viscosity in Pa s | Standard Deviation in Pa s |
|---|---|---|
| 5 | 0.24 | $2.61 \times 10^{-2}$ |
| 10 | 0.19 | $4.21 \times 10^{-2}$ |
| 20 | 0.24 | $3.13 \times 10^{-2}$ |
| 30 | 0.13 | $1.51 \times 10^{-2}$ |
| 40 | $3.76 \times 10^{-3}$ | $6.90 \times 10^{-4}$ |
| 50 | $1.23 \times 10^{-3}$ | $2.55 \times 10^{-5}$ |

**Table A4.** Viscosity values as a function of TX-100 content for a TX-100 solution without nanoparticles.

| Dispergator Content in m% | Viscosity in Pa s | Standard Deviation in Pa s |
|---|---|---|
| 5 | $6.33 \times 10^{-4}$ | $1.03 \times 10^{-5}$ |
| 10 | $6.63 \times 10^{-4}$ | $8.47 \times 10^{-6}$ |
| 20 | $7.02 \times 10^{-4}$ | $2.43 \times 10^{-5}$ |
| 30 | $7.87 \times 10^{-4}$ | $2.81 \times 10^{-6}$ |
| 40 | $8.85 \times 10^{-4}$ | $3.15 \times 10^{-6}$ |
| 50 | $9.88 \times 10^{-4}$ | $2.98 \times 10^{-6}$ |

**Table A5.** Zeta potential values as a function of PVP content for a solution with 10 m% graphite nanoparticles.

| pH Value in - | Zeta Potential | Standard Deviation in - |
|---|---|---|
| 4.2 | −21.90 | $4.08 \times 10^{-1}$ |
| 7.0 | −21.93 | $9.43 \times 10^{-2}$ |
| 7.5 | −21.47 | $4.50 \times 10^{-1}$ |
| 7.8 | −26.53 | $5.73 \times 10^{-1}$ |
| 8.0 | −30.60 | $4.03 \times 10^{-1}$ |
| 9.0 | −42.77 | $1.63 \times 10^{-1}$ |
| 9.5 | −41.22 | $1.80 \times 10^{-1}$ |
| 10.8 | −34.63 | $7.41 \times 10^{-1}$ |

**Table A6.** Zeta potential values as a function of TX-100 content for a solution with 10 m% graphite nanoparticles.

| pH Value in - | Zeta Potential | Standard Deviation in - |
|---|---|---|
| 4.2 | −16.17 | $4.03 \times 10^{-1}$ |
| 7.0 | −19.18 | $5.12 \times 10^{-1}$ |
| 8.0 | −25.17 | $4.03 \times 10^{-1}$ |
| 8.5 | −30.23 | $6.68 \times 10^{-1}$ |
| 9.0 | −27.12 | $9.27 \times 10^{-1}$ |
| 9.5 | −27.48 | $7.30 \times 10^{-1}$ |
| 10.3 | −26.93 | $2.49 \times 10^{-1}$ |
| 10.8 | −35.03 | $6.13 \times 10^{-1}$ |
| 11.0 | −34.37 | $4.92 \times 10^{-1}$ |

**Table A7.** PH values as a function of PVP content for a solution with no nanoparticles.

| Dispergator Content in m% | pH Value in - | Standard Deviation in - |
|---|---|---|
| 5 | 8.16 | $1.25 \times 10^{-2}$ |
| 10 | 8.19 | $1.63 \times 10^{-2}$ |
| 15 | 8.20 | $2.49 \times 10^{-2}$ |
| 20 | 8.22 | $1.63 \times 10^{-2}$ |
| 30 | 7.83 | $1.27 \times 10^{-1}$ |
| 40 | 7.49 | $2.49 \times 10^{-2}$ |
| 50 | 7.27 | $3.56 \times 10^{-2}$ |

**Table A8.** PH values as a function of graphite content for a solution with 20 m% PVP.

| Graphite Content in m% | pH Value in - | Standard Deviation in - |
|---|---|---|
| 5 | 4.05 | $2.16 \times 10^{-2}$ |
| 10 | 3.78 | $8.96 \times 10^{-2}$ |
| 15 | 4.12 | $7.36 \times 10^{-2}$ |
| 20 | 4.09 | $1.98 \times 10^{-1}$ |

**Table A9.** PH values as a function of TX-100 content for a solution with no nanoparticles.

| Dispergator Content in m% | pH Value in - | Standard Deviation in - |
|---|---|---|
| 5 | 7.29 | $2.87 \times 10^{-2}$ |
| 10 | 7.17 | $4.19 \times 10^{-2}$ |
| 15 | 7.22 | $3.68 \times 10^{-2}$ |
| 20 | 7.12 | $3.30 \times 10^{-2}$ |
| 30 | 7.21 | $1.63 \times 10^{-2}$ |
| 40 | 7.23 | $3.56 \times 10^{-2}$ |
| 50 | 7.21 | $2.06 \times 10^{-2}$ |

**Table A10.** PH values as a function of graphite content for a solution with 40 m% TX-100.

| Graphite Content in m% | pH Value in - | Standard Deviation in - |
|---|---|---|
| 5 | 8.82 | $5.25 \times 10^{-2}$ |
| 10 | 3.35 | $1.58 \times 10^{-2}$ |
| 15 | 3.09 | $1.30 \times 10^{-1}$ |
| 20 | 2.68 | $7.72 \times 10^{-2}$ |

**Table A11.** Viscosity values as a function of PVP content for an aqueous dispersion containing 10 m% graphite nanoparticles at a pH value of 7.8.

| Dispergator Content in m% | Viscosity in Pa s | Standard Deviation in Pa s |
|---|---|---|
| 5 | $1.54 \times 10^{-1}$ | $0.15 \times 10^{-1}$ |
| 7.5 | $0.79 \times 10^{-1}$ | $0.15 \times 10^{-1}$ |
| 10 | $0.47 \times 10^{-1}$ | $8.58 \times 10^{-3}$ |
| 12.5 | $0.13 \times 10^{-1}$ | $3.03 \times 10^{-3}$ |
| 15 | $4.07 \times 10^{-3}$ | $4.52 \times 10^{-3}$ |
| 20 | $1.30 \times 10^{-3}$ | $6.81 \times 10^{-3}$ |
| 30 | $1.18 \times 10^{-3}$ | $1.30 \times 10^{-4}$ |
| 40 | $1.92 \times 10^{-3}$ | $1.21 \times 10^{-4}$ |
| 50 | $1.26 \times 10^{-3}$ | $4.58 \times 10^{-5}$ |

**Table A12.** Viscosity values as a function of TX-100 content for an aqueous dispersion containing 10 m% graphite nanoparticles at a pH value of 10.75–11.

| Dispergator Content in m% | Viscosity in Pa s | Standard Deviation in Pa s |
|---|---|---|
| 5 | 0.18 | $3.61 \times 10^{-2}$ |
| 10 | 0.20 | $5.83 \times 10^{-2}$ |
| 20 | 0.23 | $5.56 \times 10^{-2}$ |
| 30 | 0.20 | $2.73 \times 10^{-2}$ |
| 40 | $2.71 \times 10^{-4}$ | $6.71 \times 10^{-5}$ |
| 50 | $1.25 \times 10^{-4}$ | $1.70 \times 10^{-5}$ |

**Table A13.** Particle size distribution of a dispersion containing 10 m% graphite nanoparticles and 15 m% PVP.

| Particle Size in μm | Normalized Distribution Density in μm$^{-1}$ |
|---|---|
| 1.74198 | 0.00438 |
| 1.56970 | 0.05034 |
| 1.41444 | 0.06652 |
| 1.27455 | 0.05774 |
| 1.14850 | 0.06710 |
| 1.03490 | 0.06312 |
| 0.93255 | 0.14542 |
| 0.84032 | 0.36738 |
| 0.75720 | 0.53394 |
| 0.68231 | 0.70581 |
| 0.61483 | 0.78738 |
| 0.55402 | 0.71212 |
| 0.49923 | 0.67842 |
| 0.44985 | 0.58732 |
| 0.40536 | 0.62836 |
| 0.36527 | 0.89834 |
| 0.32914 | 1 |
| 0.29659 | 0.95007 |
| 0.26726 | 0.91744 |
| 0.24082 | 0.84350 |
| 0.21700 | 0.74391 |
| 0.19554 | 0.60347 |
| 0.17620 | 0.42761 |
| 0.15877 | 0.14676 |

**Table A14.** Particle size distribution of a dispersion containing 10 m% graphite nanoparticles and 40 m% TX-100.

| Particle Size in μm | Normalized Distribution Density in $\mu m^{-1}$ |
|---|---|
| 9.21944 | $4.68477 \times 10^{-5}$ |
| 8.3076 | $3.78518 \times 10^{-4}$ |
| 7.48595 | 0.01143 |
| 6.74556 | 0.03643 |
| 6.0784 | 0.06816 |
| 5.47723 | 0.11969 |
| 4.93551 | 0.15845 |
| 4.44737 | 0.1848 |
| 4.00751 | 0.21207 |
| 3.61115 | 0.22866 |
| 3.25399 | 0.25683 |
| 2.93216 | 0.29271 |
| 2.64216 | 0.33817 |
| 2.38084 | 0.40016 |
| 2.14537 | 0.46178 |
| 1.93318 | 0.531 |
| 1.74198 | 0.59076 |
| 1.56969 | 0.63915 |
| 1.41445 | 0.67914 |
| 1.27455 | 0.69546 |
| 1.14849 | 0.72091 |
| 1.0349 | 0.74227 |
| 0.93255 | 0.7808 |
| 0.84032 | 0.84541 |
| 0.75721 | 0.90283 |
| 0.68231 | 0.97786 |
| 0.61483 | 0.99998 |
| 0.55402 | 0.97169 |
| 0.49923 | 0.87456 |
| 0.44985 | 0.6467 |
| 0.40536 | 0.4372 |
| 0.36527 | 0.16954 |
| 0.32914 | 0.01048 |
| 0.29659 | 0.00107 |

**Table A15.** Gray values of CB dispersions as a function of PVP content at different times.

| Dispergator Content in m% | Gray Value in- (as Prepared) | Gray Value in- (after 1 h) | Gray Value in- (after 12 h) |
|---|---|---|---|
| 0 | 192.31 | 194.37 | 197.1 |
| 20 | 85.97 | 87.33 | 89.73 |
| 30 | 49.31 | 51.37 | 53.41 |
| 40 | 63.97 | 65.13 | 67.89 |
| 50 | 114.39 | 119.87 | 129.13 |

**Table A16.** Gray values of CB dispersions as a function of TX-100 content at different times.

| Dispergator Content in m% | Gray Value in- (as Prepared) | Gray Value in- (after 1 h) | Gray Value in- (after 12 h) |
|---|---|---|---|
| 0 | 83.7 | 84.81 | 106.81 |
| 20 | 40.11 | 50.17 | 59.14 |
| 30 | 38.47 | 40.81 | 47.22 |
| 40 | 40.5 | 46.31 | 49.97 |
| 50 | 46.08 | 47.22 | 55.47 |

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
