# Peer review of "Improving the Dispersion Behavior of Organic Components in Water-Based Electrode Dispersions for Inkjet Printing Processes"

_applsci, doi:10.3390/app11052242_

Round 1

Reviewer 1 Report

In this work, the authors prepared graphite/carbon black dispersion for inkjet printing processes using PVP/TX-100 as the dispersant. Some properties have been studied such as viscosity, zeta potential, particle size distribution, and optical evaluation. However, the following questions should be addressed before further consideration.

  1. Can the author provide a photograph of the stable dispersion?
  2. Why pH values affect the zeta potential of the dispersion?
  3. What’s the content of each component in the dispersion? The concentration of all components should be given in weight percent (wt.%), including water.
  4. The authors measured the viscosity of the dispersion using As rheological viscosity is related to shear rate, the authors should provide the data of viscosity vs. shear rate.
  5. In Figure 5a, why the pH values decrease significantly when the content of PVP more than 20%? In Figure 6b, Why the graphite content affects the pH value of the dispersion with TX-100?
  6. The authors used NaOH and CH3COOH to adjust the pH value of the dispersion. However, NaOH and CH3COONa will become impurities after printing the electrode. How to remove NaOH and CH3COONa after printing the electrode?
  7. On page 11, the authors said that the PVP dispersant can be used as a binder. However, the minimum content of PVP was 15% or 20% in this work, which is too high as a binder in an electrode.
  8. What’s the performance of the electrode prepared with the optimal dispersion?
  9. All the Figures look weird. The authors should replot all the figures following the instructions of a scientific journal.

Author Response

Dear reviewer,

Thank you for reviewing our manuscript. We acknowledge the raised points and did our best to address all of them in a revised version of the manuscript. The changes to the original document submitted to the editor in the first place are marked in red.

On behalf of all authors yours sincerely,

Cara Greta Kolb

Reviewer 2 Report

The authors empirically studied the impact of stabilizing additives (PVP and TritonX 100) in graphite and CB dispersions and their use as electrode dispersion for inkjet printing process, with a view to their possible use as organic electrodes in lithium-ion battery (LIB). They analyze parameters such as viscosity, zeta potential, and particle size distribution.

 Applied Sciences could consider it as a potential publication.

Overall, this manuscript was quite well organized, Major comments were listed as:

One of the fundamental problems of printing technologies is the limited use of materials that can be used due to the technological limitations of the approaches used. Therefore, one point that could be improved is the section of the Introduction. In particular, unconventional jet printing techniques should be mentioned. I suggest the authors add several sentences to explain the importance of this field. Some typical references listed here should be cited.  ACS applied materials & interfaces, 2018, 10.2: 2122-2129; ACS applied materials & interfaces, 2019, 11.3: 3382-3387; Manufacturing Letters, 2014 (2), 96–99; Nature communications, 2012, 3, 890.

The considerations on additives (PVP and TX-100) have been made without considering the critical micellar concentration (CMC) typical of each surfactant.

Exceeding this value can affect parameters such as viscosity.

 Authors should add these considerations and keep them in mind in their observations and characterizations

Author Response

(The authors gave the same response as above.)

Reviewer 3 Report

This paper is focused on the stabilization of the graphite and carbon black nanoparticles in aqueous solutions.  The experimental procedure followed to produce the particle-water mixtures is complete and well described. The analysis of the dispersion properties of the mixtures takes in to account three parameters (viscosity of the mixtures, Zeta potential and the particle size distribution), in order to evaluate the presence of the sediments and the stabilization of the itself. The obtained results are well presented and discussed.

Author Response

(The authors gave the same response as above.)

Round 2

Reviewer 1 Report

If the editor accepts the figure format, I am ok with that.

Reviewer 2 Report

Thank you for making the requested revision. At the present form, I judge the manuscript as suitable for publication on Applied sciences.